# Theoretical Study on the Origin of Abnormal Regioselectivity in Ring-Opening Reaction of Hexafluoropropylene Oxide

**DOI:** 10.3390/molecules28041669

**Published:** 2023-02-09

**Authors:** Cui Yu, Yueqian Sang, Yao Li, Xiaosong Xue

**Affiliations:** 1Key Laboratory of Organofluorine Chemistry, Shanghai Institute of Organic Chemistry, University of Chinese Academy of Sciences, Chinese Academy of Sciences, 345 Lingling Road, Shanghai 200032, China; 2School of Chemistry and Materials Science, Hangzhou Institute for Advanced Study, University of Chinese Academy of Sciences, 1 Sub-lane Xiangshan, Hangzhou 310024, China

**Keywords:** ring-opening reaction, HFPO, regioselectivity, negative hyperconjugation effect

## Abstract

That nucleophiles preferentially attack at the less sterically hindered carbon of epoxides under neutral and basic conditions has been generally accepted as a fundamental rule for predicting the regioselectivity of this type of reaction. However, this rule does not hold for perfluorinated epoxides, such as hexafluoropropylene oxide (HFPO), in which nucleophiles were found to attack at the more hindered CF_3_ substituted β-C rather than the fluorine substituted α-C. In this contribution, we aim to shed light on the nature of this intriguing regioselectivity by density functional theory methods. Our calculations well reproduced the observed abnormal regioselectivities and revealed that the unusual regiochemical preference for the sterically hindered β-C of HFPO mainly arises from the lower destabilizing distortion energy needed to reach the corresponding ring-opening transition state. The higher distortion energy required for the attack of the less sterically hindered α-C results from a significant strengthening of the C(α)-O bond by the negative hyperconjugation between the lone pair of epoxide O atom and the antibonding C-F orbital.

## 1. Introduction

Epoxides play an essential role in the fields of pharmaceuticals [1], materials [2,3], and organic syntheses [4,5,6,7,8] owing to their unique biological and chemical activities. Due to the large ring strain of epoxides compared to linear aliphatic ethers, the epoxide carbon atoms can be attacked by nucleophiles to release the ring strain [9]. When the epoxide is unsymmetric, one of the two epoxide C atoms will be attacked selectively [10]. It has been well established that the less hindered carbon atom of the epoxide will be preferentially attacked under neutral and basic conditions [10,11].

Intriguingly, this rule does not apply to perfluorinated epoxides, such as hexafluoropropylene oxide (HFPO, **1**) [12]. As an important raw material for the synthesis of perfluoropolyether [13], reactions of HFPO with various nucleophiles under neutral and basic conditions have been widely investigated (Figure 1), including ring-opening (Figure 1a,b) [12,14,15,16], rearrangement (Figure 1c) [14] and anion polymerization reactions (Figure 1d) [13,17,18]. In these reactions, the nucleophile selectively attacks at the more sterically hindered CF_3_ substituted β-C instead of the fluorine substituted α-C.

Several explanations have been proposed for the anomalous ring-opening regioselectivity of HFPO. Sianesi et al. [14] hypothesized that nucleophiles preferentially attack at the more hindered CF_3_ substituted β-C atom due to its higher electrophilicity endowed by the strong electron-withdrawing trifluoromethyl group. Besides the increased electrophilicity of the β-C atom by the trifluoromethyl group, Eleuterio [13] proposed that the lone pair repulsion between fluorine on the α-C atom and the incoming nucleophiles also contributes to the observed abnormal regioselectivity. Jiang et al. [19,20] attributed the abnormal regioselectivity to the strengthening of the C(α)-O bond by the α-fluorine effect [21,22]. Lemal and Sudharsanam [23] put forward that negative hyperconjugation [24,25,26,27,28,29,30] of the developing oxyanion stabilizing the ring-opening transition state was at least one cause. 

To date, the origin of abnormal ring-opening regioselectivity of HFPO remains unclear. As a continuation of our research efforts on the structure−reactivity relationship and mechanism of fluorination/fluoroalkylation reagents [31,32,33,34,35,36,37,38,39,40,41,42,43,44,45,46,47,48], we investigated the mechanism of the ring-opening reaction of HFPO by density function theory (DFT) calculations and performed distortion/interaction activation strain analysis [49,50,51] and Natural Bond Orbital (NBO) [52,53] analysis to explore the determinant of the abnormal regioselectivity.

## 2. Results and Discussion

The ring-opening reactions of HFPO (**1**) and propylene oxide (PO, **2**) by fluoride were employed as the model reactions (Figure 2). The computed Gibbs free energy profiles for the two ring-opening reactions are shown in Figure 1. In the reaction of **1** (Figure 1a), nucleophilic attack at the more sterically encumbered CF_3_ substituted β-C (β-attack) by fluoride has an activation free energy barrier of only 7.6 kcal/mol via **TS1β** to give **Int1β**. This process is highly exergonic and irreversible. Then, the release of fluoride from the **Int1β** gives perfluoropropionic acid fluoride **P1β**. Nucleophilic attack at the α-C (α-attack) has a higher barrier of 11.5 kcal/mol, leading to hexafluoroacetone **P1α** after the exclusion of fluoride. Accordingly, the attack at the more sterically hindered CF_3_ substituted β-C is preferred over the attack at the α-C by 3.9 kcal/mol in the ring-opening reaction of **1**. Conversely, nucleophilic attack at the α-C (**TS2α**) is favored over the β-C (**TS2β**) by 2.9 kcal/mol in the ring-opening reaction of **2** (Figure 1b). Substantially higher barriers are required for ring-opening reaction of **2** than of **1**. Overall, the predicted regioselectivities and reactivities are consistent with the experimental results [12].

To reveal the physical factors determining the distinct regioselectivity, we performed distortion/interaction activation strain analysis (DIAS) for the two model reactions (Figure 2a,b). DIAS analysis revealed that the preferred nucleophilic attack at the more sterically hindered CF_3_ substituted β-C of **1** originates mainly from the lower destabilizing distortion energy (Figure 2a). The stabilizing interaction energy for β-attack is slightly weaker than for α-attack.

The lower destabilizing distortion energy for β-attack in the ring-opening of **1** can be attributed to the weaker C(β)-O bond strength relative to C(α)-O bond as indicated by their bond lengths (C(α)-O bond length: 1.374 Å vs. C(β)-O bond length: 1.415 Å) [54]. Substitution of all H-atoms in **2** with F enhances the strength of both C(β)-O and C(α)-O bonds (C(α/β)-O: 1.431/1.432 Å in **2** vs. 1.374/1.415 Å in **1**). We noted, however, that the C(α)-O bond was strengthened even more. This can be mainly ascribed to the stronger negative hyperconjugation (Figure 2c) between the lone pair of O atom and the antibonding C-F orbital compared to the antibonding C-CF_3_ orbital (n(O) → σ*_C-F_ is at least 9.2 kcal/mol more stable than n(O) → σ*_C- CF3_) [24,25,26,27,28,29,30], owing to the former is a better electron-acceptor (the electron-acceptor capability: σ*_C-F_ > σ*_C- CF3_/σ*_C-H_). The n(O) → σ*_C-F_ negative hyperconjugation strengthens the C-O bond and weakens the C-F bond. Notably, as the negative charge on the O atom accumulates during the ring-opening, the stabilizing stereo-electronic interactions between oxygen lone pairs and antibonding C-F orbitals become even stronger in the transition states, thus causing an even larger difference in distortion energy around the transition state zone of the ring-opening of **1** (Figure 2a and Appendix A).

The calculated electrophilicity indexes [55,56] indicate that the β-C is more electrophilic than the α-C in **1** (Figure 2d and Appendix A). Moreover, the antibonding C(β)-O orbital is lower in energy than the antibonding C(α)-O orbital (Figure 2e and Appendix A). Irrespective of steric repulsion, β-attack in the ring-opening of **1** should exhibit stronger interaction energy (electrostatic and orbital interactions). Thus, the observed slightly weaker interaction energy for β-attack must result from a more significant destabilizing steric repulsion between the incoming fluoride and the trifluoromethyl group (Figure 2f) that overrides the stabilizing electrostatic and orbital interactions. Indeed, due to the destabilizing steric repulsion, the F^−^···β-C distance is longer in TS1β than in TS1α, leading to a loss of stabilizing orbital and electrostatic interactions. 

Consistent with the previous report of Hamlin, Bickelhaupt, and coworkers [57], the α-C regioselectivity of **2** is controlled by interaction energy (Figure 2b) that mainly originates from the lower steric repulsion (Figure 2f). The lower steric repulsion of α-C allows the development of stronger stabilizing orbital and electrostatic interaction between the incoming nucleophile and this position. The α-C is more electrophilic than β-C in **2** (Figure 2d and Appendix A), and the antibonding C(α)-O orbital is lower in energy than the antibonding C(β)-O orbital (Figure 2e and Appendix A), owing to the electron-donating of the methyl group. The strengths of C(β)-O and C(α)-O bonds are nearly identical (C(α)-O bond length: 1.431 Å vs. C(β)-O bond length: 1.432 Å), and thus the destabilizing distortion energies are similar in magnitude for β- and α-attack.

The higher ring-opening reactivity of **1** can be attributed to (1) the effective stabilizing of the developing negative charges on the O atom in transition structure by negative hyperconjugation (n(O) → σ*C-F) and strong electron-withdrawing capability of F/CF_3_ and (2) the enhanced HOMO (F^−^)-LUMO (substrate) orbital interaction as a result of the LUMO lowering after fluorine substitution.

## 3. Computational Details

DFT calculations were performed with the Gaussian 16 programs [58]. Geometry optimization and frequency calculations were carried out at the ꞷB97X-D/6-31+G(d,p)-SMD-(ethyl ether) level of theory [59,60,61,62,63,64,65,66]. All the transition states have only one imaginary frequency, while the stationary points do not. Intrinsic reaction coordinates (IRC) calculations at the same level verified the connectivity of located intermediates and transition states. Single-point energy was calculated at the level of ꞷB97X-D/6-311++G(2df,2p)-SMD-(ethyl ether) [67]. Non-covalent interaction (NCI) analysis was performed by NCIplot 3.0 [68]. Wavefunctions analysis and reactivity descriptors calculation were performed by Multiwfn 3.8 [69,70], for which substrates were reoptimized at the ꞷB97X-D/6-31G(d,p)-SMD-(ethyl ether) level. Distortion/interaction activation strain analysis (DIAS) was realized by a python tool named autoDIAS [71]. Figures of NCI and orbital isosurfaces were plotted with PyMOL 2.0.4 [72]. The 3D plots for the molecules were created with CYLview 20 [73].

Distortion/interaction activation strain analysis is a popular model [49,50,51] developed to investigate the determinant of chemical reactivity. In this model, the total energy increased along the reaction pathway is decomposed into the distortion energy of substrates and the interaction energy between the two reactants (1).
(1)$$\Delta E_{total}=\Delta E_{interaction}+\Delta E_{distortion}$$

The distortion energy describes the energy needed to deform the substrate from the equilibrium structure to deformed structure along the reaction coordinate (2). The interaction energy refers to the energy all the interaction between the two deformed reactants (3).
(2)$$\Delta E_{distortion}=\Delta E_{distored-structure}-\Delta E_{equilibrium-structure}\;$$
(3)$$\Delta E_{interaction}=\Delta E_{total}-\Delta E_{distortion\;}$$

## 4. Conclusions

In conclusion, we revealed that the abnormal regioselectivity of the ring-opening reaction of HFPO originates from a significant strengthening of C-O bond by the negative hyperconjugation between the lone pair of epoxide O atom and the antibonding C-F orbital, which leads to the higher destabilizing distortion energy needed to reach the ring-opening transition state for attacking at the less sterically hindered carbon. Our results provide strong support for Jiang’s [20] and Lemal’s [23] hypotheses. We expected that the insights from this study would enrich our understanding of unique fluorine effects in organic reactions [25,74,75,76,77,78] and should be helpful for the design of new regioselective ring-opening reactions of epoxides by regulating the electronic properties of the substituents.

## Data Availability

Not applicable.

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
