# Peer review of "Theoretical Study on the Origin of Abnormal Regioselectivity in Ring-Opening Reaction of Hexafluoropropylene Oxide"

_molecules, 2023, doi:10.3390/molecules28041669_

Round 1

Reviewer 1 Report

In general, this is excellent theoretical work. The topic is interesting, and the applied level of theory is adequate. I recommend the publication of this paper after  consideration os issues listed below:
+ Application of the êž·B97X-D functional for the exploration of different type organic reactions should be supported by respective references
+ Figure 1:
The intermediate exhibit relative energy about 10okcal/mol lower than the final product. This is a not error? Maybe the energy of the final state was calculated without fluorine anion? This should be checked and corrected.
+ Application of the FMO theory for the interpretation of the organic reaction course is completly outdated. So, respective figures and paragraphs in the text should be removed.

Author Response

In general, this is excellent theoretical work. The topic is interesting, and the applied level of theory is adequate.

Reply: We appreciate the reviewer’s positive evaluation of our work and efforts in reviewing our manuscript. We have revised the manuscript accordingly. Our point-by-point responses are detailed below.

  1. Application of the êž·B97X-D functional for the exploration of different type organic reactions should be supported by respective references

Reply: Thanks for this reviewer’s good suggestion. We added three references [60-62] to support the êž·B97X-D functionals selected in this work.

  1. Galabov, B.; Koleva, G.; Schaefer, H. F., 3rd; Allen, W. D., Nucleophilic Influences and Origin of the S(N)2 Allylic Effect. Chem. Eur. J. 2018, 24, 11637-11648.
  2. Grimme, S.; Antony, J.; Ehrlich, S.; Krieg, H., A consistent and accurate ab initio parametrization of density functional dispersion correction (DFT-D) for the 94 elements H-Pu. J. Chem. Phys. 2010, 132, 154104.
  3. Gonzales, J. M.; Cox, R. S.; Brown, S. T.; Allen, W. D.; Schaefer, H. F., Assessment of Density Functional Theory for Model SN2 Reactions:  CH3X + F- (X = F, Cl, CN, OH, SH, NH2, PH2). J. Phys. Chem. A 2001, 105, 11327-11346.

  1. Figure 1: The intermediate exhibit relative energy about 10 kcal/mol lower than the final product. This is a not error? Maybe the energy of the final state was calculated without fluorine anion? This should be checked and corrected.

Reply: Thanks for the review’s carefulness. We carefully checked our results and found they were right. Similar results have been reported by Allen (Chem. Eur. J. 2018, 24, 11637-11648). The last step is endergonic due to the favorable interaction between the product and anion in the intermediate.

  1. Application of the FMO theory for the interpretation of the organic reaction course is completly outdated. So, respective figures and paragraphs in the text should be removed.

Reply: We agree that the FMO theory is a classical and powerful tool for understanding the activity of organic reactions.  Since it is very helpful for understanding the negative hyperconjugation, we decided to keep them in the manuscript.

Reviewer 2 Report

The manuscript by Yu et al. studies the regioselectivity in ring-opening reactions of hexafluoropropylene oxide (HFPO), where nucleophiles were found to attack at the more hindered CF3 substituted β-C rather than the fluorine substituted α-C, by means of density functional theory-based calculations.

The simulations were performed at the \{omega}B97X-D/6-31+G(d,p)-SMD-(ethyl ether) level of theory through the Gaussian 16 code and the analyses included intrinsic reaction coordinates (IRC), non-covalent interaction (NCI) with NCIplot 3.0 and PyMOL 2.0.4, distortion/interaction activation strain analysis (DIAS) by means of autoDIAS, and others.

The ring-opening reactions of HFPO (1) and propylene oxide (PO, 2) by fluoride were employed as the model reactions. The calculations reproduced the observed regioselectivity of HFPO and the Authors concluded the preference for the attack of the fluoride at the sterically hindered β-C of HFPO mainly arises from the lower destabilizing distortion energy needed to reach the corresponding ring-opening transition state, while the higher distortion energy required for the attack at the less sterically hindered α-C, resulting from a significant strengthening of the C(α)-O bond by the negative hyperconjugation between the lone pair of epoxide O atom and the antibonding C-F orbital, renders it less preferred.

The manuscript is concise and quite understandably written, apart from a few parts I will mention in the comments below. The Authors simulate the ring-opening reaction of HFPO (1) by fluoride and compare it with that of PO (2) under the same conditions. The methodologies they use in the analysis, although less extensive, less characterized, and less rationalized, are along the lines of those carried out in Ref. [70]. Some things need to be clarified, including some major fundamental aspects as the rationale behind the choice of the model reactions taken into considerations and others I will detail in my comments below. I did not have access to the Supplementary Materials, but from the description it should not be fundamental to follow the arguments and reasoning of the Authors.

In the present form I doubt the manuscript has the impact required for publication in Molecules, but I believe it could represent a nice addition to the literature for regioselectivity in ring-opening reactions of HFPO, after the Authors have addressed my comments and worked on the text of the manuscript.

1. How did the Authors select the model reactions considered? In particular, given the well-known strong dependence of the reaction path on the basic vs acidic reaction conditions, it doesn't seem the Author considered different conditions in the analysis. My interpretation is that both the reactions considered are mainly in acidic conditions, which result in the expected behavior for the PO molecule (2) and the behaviour of HFPO, target of the analysis. I believe the rationale behind these choices and what is known, what is expected and what is an original finding of this work needs to be clarified and expanded in the text. Did the Author have some indication to select these conditions? Would different conditions modify the findings and how extensively?

2. How did the Authors select the implicit solvent used in the calculations? Since it has been shown the ring-opening reactions of epoxides is strongly dependent on the reaction conditions - i.e., basic vs acidic - it is an important point, even though of course the solvent does not interact chemically with the molecules in the simulations being described through a polarizable continuum model (PCM). Can the Authors expand on the computational details of the PCM approach adopted?

3. At some point in the text the Authors refer to calculations including a substrate. This is not clear either in the results or in the discussion  or in the computational details. In particular there are no results shown I can relate to that. Please clarify.

4. Pag. 5 "... the antibonding C(β)-O orbital is lower in energy than the antibonding C(α)-O orbitals ...". Discussing the absolute energy position of the LUMO orbital without referring it to the HOMO or discussing the electronic structures of the molecules makes absolutely no sense. Please extend the analysis and the discussion.

5. Pag. 5 the paragraph beginning with "This can be mainly ascribed...". I cannot understand this argument on the negative hyperconjugation in connection with the orbital transitions as it finds no clear support in the presented results and it's difficult to follow. What does it mean "...the former is a better acceptor..."?

6. I assume the red and black dots in Figs. 2a,b refer to the "equilibrium" C-O bond distance in the molecule along the S_N2 pathway, but I don't understand how the values correlate with the C-O bond length discussed in the text at the beginning of pag. 5.

7. What is the the transition state zone the Authors refer to in Figs. 2a and S1? 

8. I see from the Computational details section the Authors did check that the calculated geometries corresponding to stationary points are actual energy minima, i.e., no imaginary frequencies in vibrational analysis, and similarly for the transition states (one imaginary frequency). How were the vibrational analyses performed?

9. Pag. 5 "Both suggest, irrespective of steric repulsion ... loss of stabilizing orbital and electrostatic interactions". I don't understand the reasoning here.

Typos and others:

- Many typos refers to "the attack at" missing the "at" or similar;

- Abstract: "That nucleophiles preferentially attack the less..." --> "That nucleophiles preferentially attack at the less...";

- Abstract: "...nucleophiles were found to attack the more..." --> "...nucleophiles were found to attack at the more...";

- Pag. 4, first paragraph: "1" should be bold;

- Conclusion: "...for attacking the less sterically hindered carbon..." --> "...for attacking at the less sterically hindered carbon...";

- Conclusion: "[71-76] should be helpful" --> "[71-76] and should be helpful";

- Conclusion: "...would enriches..." --> "...would enrich...".

Author Response

Reviwer 2:

  1. “How did the Authors select the model reactions considered? In particular, given the well-known strong dependence of the reaction path on the basic vs acidic reaction conditions, it doesn't seem the Author considered different conditions in the analysis. My interpretation is that both the reactions considered are mainly in acidic conditions, which result in the expected behavior for the PO molecule (2) and the behavior of HFPO, target of the analysis. I believe the rationale behind these choices and what is known, what is expected and what is an original finding of this work needs to be clarified and expanded in the text. Did the Author have some indication to select these conditions? Would different conditions modify the findings and how extensively?”

Reply: Thanks for the reviewer’s comments. HFPO is an important raw material, and the reactions of HFPO with various nucleophiles are mostly under neutral and basic conditions (Scheme 1), including ring-opening (Scheme 1a,b), rearrangement (Scheme 1c) and anion polymerization reactions (Scheme 1d). In these reactions, the nucleophile selectively attacks at the more sterically hindered CF3 substituted β-C instead of the fluorine substituted α-C. Our study mainly focused on the abnormal regioselectivity of the ring-opening reaction of HFPO under neutral and basic conditions. And fluorine ion is one of the most popular nucleophiles. Thus we selected the reaction of fluoride and HFPO under neutral and basic conditions as a model reaction to study the origin of the abnormal regioselectivity. We studied the reaction of PO with fluoride for comparison. Our calculations well reproduced the observed abnormal regioselectivities and revealed that the unusual regiochemical preference for the sterically hindered β-C of HFPO mainly arises from the lower destabilizing distortion energy needed to reach the corresponding ring-opening transition state. The higher distortion energy required for the attack of the less sterically hindered α-C results from a significant strengthening of the C(α)-O bond by the negative hyperconjugation between the lone pair of epoxide O atom and the antibonding C-F orbital. We expected that the insights from this study would enrich our understanding of unique fluorine effects in organic reactions and should be helpful for the design of new regioselective ring-opening reactions of epoxides, under neutral and basic conditions, by regulating the electronic properties of the substituents. A comprehensive study of different conditions (acid conditions) will be conducted in our future work.

  1. “How did the Authors select the implicit solvent used in the calculations? Since it has been shown the ring-opening reactions of epoxides are strongly dependent on the reaction conditions - i.e., basic vs acidic - it is an important point, even though of course the solvent does not interact chemically with the molecules in the simulations being described through a polarizable continuum model (PCM). Can the Authors expand on the computational details of the PCM approach adopted?”

Reply: In the present study, we selected Truhlar’s SMD implicit solvent model [59] for all calculations. The SMD has been widely used to account for the solvent effects with reasonable accuracy. The use of PCM sometimes suffers the problem of failure of convergence during structural optimization.

  1. “At some point in the text the Authors refer to calculations including a substrate. This is not clear either in the results or in the discussion or in the computational details. In particular, there are no results shown I can relate to that. Please clarify.”

Reply: We have clarified the following descriptions: page 5: “The lower destabilizing distortion energy for β-attack in the ring-opening of 1 can be attributed to the weaker C(β)-O bond strength relative to C(α)-O bond as indicated by their bond lengths (C(α)-O bond length: 1.374 Å vs C(β)-O bond length: 1.415 Å)”. “Notably, as the negative charge on the O atom accumulates during the ring-opening, the stabilizing stereo-electronic interactions between oxygen lone pairs and antibonding C-F orbitals become even stronger in the transition states, thus causing an even larger difference in distortion energy around the transition state zone of the ring-opening of 1.”,  “Both suggest that β-attack in the ring-opening of 1 should exhibit stronger interaction energy.”

  1. “Pag. 5 "... the antibonding C(β)-O orbital is lower in energy than the antibonding C(α)-O orbitals ...". Discussing the absolute energy position of the LUMO orbital without referring it to the HOMO or discussing the electronic structures of the molecules makes absolutely no sense. Please extend the analysis and the discussion.”

Reply: Due to the nucleophile in the two model reactions being the same, the comparison of LUMOs (their relative energy) electrophiles would be informative. The anti-bonding orbital energy discussed in the main text is also relative energy. The lower anti-bonding orbital energy or LUMO indicates a better electron acceptor.

  1. “Pag. 5 the paragraph beginning with "This can be mainly ascribed...". I cannot understand this argument on the negative hyperconjugation in connection with the orbital transitions as it finds no clear support in the presented results and it's difficult to follow. What does it mean "...the former is a better acceptor..."?”

Reply: Negative hyperconjugation refers to the donation of electron density from a filled – or p-orbital to a nearby σ*-orbital. The lower the energy of the antibonding orbital is, the greater the electron-acceptor ability the bond has [Eric V.Anslyn. Modern Physical Organic Chemistry. University Science: America. 2005;].

As shown in Figure 2c, the value of E(2) Second Order Perturbation Energy also supports this conclusion. The value of E(2) of n(O) → σ*C- CF3 is only 7.4 kcal/mol, while the value of E(2) of n(O) → σ*C- F is 16.6 kcal/mol.

  1. “I assume the red and black dots in Figs. 2a,b refer to the "equilibrium" C-O bond distance in the molecule along the S_N2 pathway, but I don't understand how the values correlate with the C-O bond length discussed in the text at the beginning of pag. 5.”

Reply: Figures 2a,b are the results of DIAS for the reactions of HFPO and PO, respectively. The electronic energy values are projected on the distance between the broken C-O bond along the reaction coordinate. The dots marked on the curve represent the transition state of the reaction. We have added “(positions of TS indicated with a dot)” in the Annotations of the two figures.

  1. “What is the transition state zone the Authors refer to in Figs. 2a and S1?”

Reply: The dots marked on the curve represent the transition state of the reaction. We have added “(positions of TS indicated with a dot)” in the Annotations of the two figures.

  1. “I see from the Computational details section the Authors did check that the calculated geometries corresponding to stationary points are actual energy minima, i.e., no imaginary frequencies in vibrational analysis, and similarly for the transition states (one imaginary frequency). How were the vibrational analyses performed?”

Reply: The results of the vibrational analysis were performed by Gaussview. With Gaussview we check and analyze whether the vibrational motion corresponding to the imaginary frequency of the ring-opening.

  1. “Pag. 5 "Both suggest, irrespective of steric repulsion ... loss of stabilizing orbital and electrostatic interactions". I don't understand the reasoning here.”

Reply: ∆Eint is usually decomposed into three parts, electrostatic interaction energy, Pauli repulsion energy, and orbital interaction energy between the epoxide and fluoride. As shown in Figure 2a, the ∆Eint of the transition state of the 1-α pathway is slightly stronger than that of 1-β pathway. However, electrophilic indexes (the larger the value is, the more electrophilic the position is) and relative orbital energy indicated the ∆Eint of the transition state of the 1-α pathway should be weaker than that of 1-β pathway irrespective of steric repulsion. Thus, the destabilizing steric repulsion of TS1β is stronger than that of TS1α, which was investigated and supported by NCI analysis. We have revised the sentences as “Both suggest that β-attack in the ring-opening of 1should exhibit stronger interaction energy.”

  1. Typos and others:”

- Many typos refers to "the attack at" missing the "at" or similar;

Abstract: "That nucleophiles preferentially attack the less..." --> "That nucleophiles preferentially attack at the less...";

Abstract: "...nucleophiles were found to attack the more..." --> "...nucleophiles were found to attack at the more...";

Pag. 4, first paragraph: "1" should be bold;

Conclusion: "...for attacking the less sterically hi

Reply: Thanks for your carefulness. We have corrected all typos.

Reviewer 3 Report

Dear Authors,

I have enjoyed reading this concise theoretical study. This work shows the use of electronic structure theory to demystify unusual chemical reactivity. The computational methods used for this study are adequate for a system under investigation and all the results are convincingly demonstrated. Moreover, use of different tools such as DFT, NCI, NBO, DIAS makes the study very comprehensive.

Author Response

We appreciate the reviewer’s positive evaluation of our work.

Round 2

Reviewer 2 Report

The manuscript by Yu et al. studies the regioselectivity in ring-opening reactions of hexafluoropropylene oxide (HFPO), by means of density functional theory-based calculations.

I went through the new version of the manuscript. The Authors addressed the comments in my referee report and for some of them they modified the manuscript accordingly, even though not to a large extent. I still believe some more work on the text would make it more easily understandable for the average reader and that some of my points – such as points 1, 2, 4 and 5 for instance – would have deserved some more consideration and/or more important modification of the text.

Anyway, I believe in this form the manuscript represents a nice addition to the literature for regioselectivity in ring-opening reactions of HFPO and therefore, my suggestion is to publish the manuscript in Molecules, after the Authors have re-considered some of the points in my previous report and the relative response for the text of the manuscript. I will not need to see the manuscript again.

Author Response

Reviewer 2 (Round 2):

The manuscript by Yu et al. studies the regioselectivity in ring-opening reactions of hexafluoropropylene oxide (HFPO), by means of density functional theory-based calculations.

I went through the new version of the manuscript. The Authors addressed the comments in my referee report and for some of them they modified the manuscript accordingly, even though not to a large extent. I still believe some more work on the text would make it more easily understandable for the average reader and that some of my points – such as points 1, 2, 4 and 5 for instance – would have deserved some more consideration and/or more important modification of the text.

Anyway, I believe in this form the manuscript represents a nice addition to the literature for regioselectivity in ring-opening reactions of HFPO and therefore, my suggestion is to publish the manuscript in Molecules, after the Authors have re-considered some of the points in my previous report and the relative response for the text of the manuscript. I will not need to see the manuscript again.

Reply: Thanks  once again for your comments and suggestions. The following is our updated response to your first round of comments.

  1. “How did the Authors select the model reactions considered? In particular, given the well-known strong dependence of the reaction path on the basic vs acidic reaction conditions, it doesn't seem the Author considered different conditions in the analysis. My interpretation is that both the reactions considered are mainly in acidic conditions, which result in the expected behavior for the PO molecule (2) and the behavior of HFPO, target of the analysis. I believe the rationale behind these choices and what is known, what is expected and what is an original finding of this work needs to be clarified and expanded in the text. Did the Author have some indication to select these conditions? Would different conditions modify the findings and how extensively?”

Reply: Thanks for the reviewer’s comments. HFPO is an important raw material, and the reactions of HFPO with various nucleophiles are mostly under neutral and basic conditions (Please see: Millauer, H.; Schwertfeger, W.; Siegemund, G., Hexafluoropropene Oxide — A Key Compound in Organofluorine Chemistry. Angew. Chem. Int. Ed. 1985, 24, 161-179.). In these reactions, the nucleophile selectively attacks at the more sterically hindered CF3 substituted β-C instead of the fluorine substituted α-C. Thus, our study mainly focused on the abnormal regioselectivity of the ring-opening reaction of HFPO under neutral and basic conditions. And fluorine ion is one of the most popular nucleophiles. Accordingly, we selected the reaction of fluoride and HFPO under neutral and basic conditions as a model reaction to study the origin of the abnormal regioselectivity. We studied the reaction of PO with fluoride for comparison.

We believe that a comprehensive study of different conditions (acidic and basic conditions) is of great importance but should be another project.

  1. “How did the Authors select the implicit solvent used in the calculations? Since it has been shown the ring-opening reactions of epoxides are strongly dependent on the reaction conditions - i.e., basic vs acidic - it is an important point, even though of course the solvent does not interact chemically with the molecules in the simulations being described through a polarizable continuum model (PCM). Can the Authors expand on the computational details of the PCM approach adopted?”

Reply: It should be noted that the reactions of HFPO with various nucleophiles are mostly under neutral and basic conditions (Please see: Millauer, H.; Schwertfeger, W.; Siegemund, G., Hexafluoropropene Oxide — A Key Compound in Organofluorine Chemistry. Angew. Chem. Int. Ed. 1985, 24, 161-179.). In the present study, we selected Truhlar’s SMD implicit solvent model for all calculations. The SMD has been widely used to account for the solvent effects with reasonable accuracy(Cheong, P. H. Y.; Legault, C. Y.; Um, J. M.; Celebi-Olcum, N.; Houk, K. N., Quantum Mechanical Investigations of Organocatalysis: Mechanisms, Reactivities, and Selectivities. Chem. Rev. 2011, 111, 5042-5137.). The use of PCM sometimes suffers the problem of failure of convergence during structural optimization.

  1. “Pag. 5 "... the antibonding C(β)-O orbital is lower in energy than the antibonding C(α)-O orbitals ...". Discussing the absolute energy position of the LUMO orbital without referring it to the HOMO or discussing the electronic structures of the molecules makes absolutely no sense. Please extend the analysis and the discussion.”

Reply: Due to the nucleophile in the two model reactions being the same, the comparison of LUMOs (their relative energy) electrophiles would be informative. The anti-bonding orbital energy discussed in the main text is also relative energy. The lower anti-bonding orbital energy or LUMO indicates a better electron acceptor.

  1. “Pag. 5 the paragraph beginning with "This can be mainly ascribed...". I cannot understand this argument on the negative hyperconjugation in connection with the orbital transitions as it finds no clear support in the presented results and it's difficult to follow. What does it mean "...the former is a better acceptor..."?”

Reply: Negative hyperconjugation refers to the donation of electron density from a filled – or p-orbital to a nearby σ*-orbital. The lower the energy of the antibonding orbital is, the greater the electron-acceptor ability the bond has [Eric V.Anslyn. Modern Physical Organic Chemistry. University Science: America. 2005;].

As shown in Figure 2c, the value of E(2) Second Order Perturbation Energy also supports this conclusion. The value of E(2) of n(O) → σ*C- CF3 is only 7.4 kcal/mol, while the value of E(2) of n(O) → σ*C- F is 16.6 kcal/mol.
